# Impact of Coastal Infrastructure on Ocean Colour Remote Sensing: A Case Study in Jiaozhou Bay, China

**Yuan Yuan** [1,2,*] **, Isabel Jalón-Rojas** [1,2] **and Xiao Hua Wang** [1,2]

1   School of Science, The University of New South Wales, Canberra ACT 2600, Australia;
    i.jalonrojas@adfa.edu.au (I.J.-R.); x.h.wang@unsw.edu.au (X.H.W.)
2   The Sino-Australian Research Centre for Coastal Management, The University of New South Wales,
    Canberra ACT 2600, Australia
*   Correspondence: yuan.yuan1@student.adfa.edu.au; Tel.: +61-0416341127

**Abstract:** Spatial and temporal ocean colour data are increasingly accessible through remote sensing, which is a key tool for evaluating coastal biogeochemical and physical processes, and for monitoring water quality. Coastal infrastructure such as cross-sea bridges may impact ocean colour remote sensing due to the different spectral characteristics of asphalt and the seawater surface. However, this potential impact is typically ignored during data post-processing. In this study, we use Jiaozhou Bay (East China) and its cross-bay bridge to examine the impact of coastal infrastructure on water-quality remote-sensing products, in particular on chlorophyll-a concentration and total suspended sediment. The values of these products in the bridge area were significantly different to those in the adjacent water. Analysis of the remote-sensing reflectance and application of the Normalised Difference Water Index demonstrate that this phenomenon is caused by contamination of the signal by bridge pixels. The Moderate Resolution Imaging Spectroradiometer (MODIS) products helped estimate the pixel scale that could be influenced by contamination. Furthermore, we found similar pixel contamination at Hangzhou Bay Bridge, suggesting that the impact of large coastal infrastructure on ocean colour data is common, and must therefore be considered in data post-processing.

**Keywords:** coastal infrastructure; remote sensing; data quality; Jiaozhou Bay; GOCI; chlorophyll-a; total suspended sediment

## 1. Introduction

Ocean colour measurements contribute significantly to coastal ecosystem restoration, monitoring of dredging and dumping, fisheries management and a wide variety of research on water quality and coastal biogeochemical and physical processes. Among the ocean colour variables available from remote-sensing data, chlorophyll-a concentration (chl-a), total suspended sediment (TSS) and coloured dissolved organic matter (CDOM) are usually of concern in water quality assessment [1,2]; chl-a, primary productivity and red tide index are more important in studies on biogeochemical processes [3,4]; TSS and the diffuse attenuation coefficient at 490 nm (Kd490) may be included in evaluation of physical processes [5,6].

The Geostationary Ocean Colour Imager (GOCI) was the first geostationary ocean colour sensor, launched by South Korea in June 2010. GOCI covers 2500 km × 2500 km of the northeast Asian region, with a spatial resolution of 500 m and a temporal resolution of 1 h. GOCI retrieves remote-sensing reflectance, chl-a and TSS. The values of these parameters have been validated by in situ data along the Korean coast, showing a relatively good agreement for remote-sensing reflectance (Rrs) except band 1 [7,8], but not satisfactory for chl-a using the ocean chlorophyll 2 algorithm (OC2) [8].

Coastal infrastructure for urban development has dramatically increased over the last few decades. Cross-sea bridges connect coasts and can extend tens of kilometres over coastal waters. The Jiaozhou Bay (JZB) Bridge, the Hangzhou Bay Bridge and the newly built Hong Kong–Zhuhai–Macau Bridge are examples of bridges crossing regional seas. The spectral characteristics of road-bridge surfaces are different from the sea surface, which may result in errors in some ocean colour products. For terrestrial remote sensing, many efforts have been made to improve the algorithms to detect and classify different types of land cover (e.g., trees, grass, blocks) in hyperspectral imaging data [9]. The different spectral responses of each material enable extraction of one expected land cover, with others discarded. For example, when vegetation is expected, the water, soil, biomass-burning smoke and other land covers should be excluded [10]. For the oceans, although cross-sea bridges exist worldwide, bridge impacts on ocean colour remote sensing are typically ignored in data post-processing and discussion. For example, Gao et al. [11] validated their sediment model for JZB using the GOCI TSS product without paying attention to potential bridge contamination. Hangzhou Bay Bridge was also not considered in a GOCI-based assessment of the diurnal variation in chl-a and TSS [12], nor in mapping the suspended particulate matter [13].

In this short communication, a case study of the impact of Jiaozhou Bay Bridge on GOCI products is described to illustrate, for the first time, the bridge effects on ocean colour remote sensing. The ocean environment of Jiaozhou Bay is first described in Section 2, and the GOCI products and the methodology used in this short communication in Section 3. Finally, the potential contamination by coastal infrastructure of GOCI products is discussed in Section 4.

## 2. Study Area

Located on the coast of the Yellow Sea (Eastern China, Figure 1a), Jiaozhou Bay (JZB) is included in the GOCI target area. This shallow semi-enclosed bay is surrounded by Qingdao City (Figure 1b), and is 33 km long and 28 km wide. The average water depth is 7 m, with the greatest depth of 64 m in the middle channel.

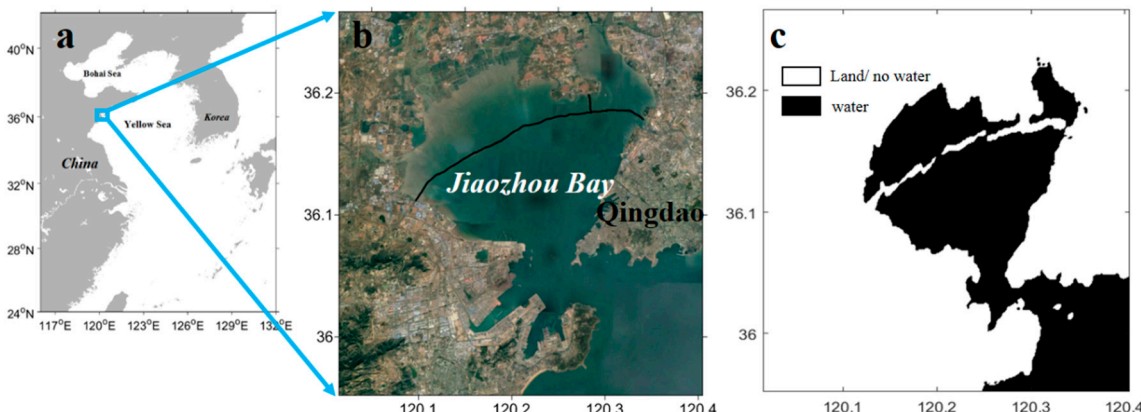

**Figure 1.** (**a**) Location of Jiaozhou Bay (JZB) on the Yellow Sea coast. (**b**) Satellite image of JZB from Google Earth (provided by DigitalGlobal); the black line shows the JZB Bridge. (**c**) The normalised difference water index distribution for JZB (derived from the Geostationary Ocean Colour Imager).

JZB is a typical eutrophic ecosystem characterized by high nitrogen and phosphorus concentrations. Monthly mean chl-a fluctuated between 0.99 and 18.67 µg/L from March 2005 to February 2006, with the maximum values and largest fluctuations in August [14]. Chlorophyll-a concentration is higher in the northeast and northwest of the bay, with a gradual decrease towards the south [15]. The circulation in JZB is dominated by tidal currents, with a minimum speed of 0.15 m/s [16]. The maximum currents occur at the entrance of the bay, with a magnitude of 1.7 m/s [16]. Tides are semidiurnal; among the tidal constituents, $M_2$ is the most predominant [17]. The residual currents characterize a weak clockwise circulation in the shallow regions and a strong clockwise eddy in the southern part of

JZB [18]. The clockwise circulation tends to be changed by the southeasterly wind (happens in spring and summer) to a relatively strong northwestward water transport near the surface [18]. The Dagu River is the major source of sediments, accounting for over 64% of the total sediment input, based on the annual averages between 1960 and 2008 [19]. The net suspended-sediment transport in JZB is directed towards the mouth of the bay, and is of the order of $10^3$ tonnes per tidal cycle [20,21]. The suspended-sediment concentrations are high (10–50 mg/L) in the northwest and low (<10 mg/L) in the east [21]. JZB has also experienced a large increase in pollutants during the last three decades [22], which has caused the water quality to deteriorate [23].

From 2007 to 2011, a 27 km-long road bridge (called here the JZB Bridge) was built across JZB to link parts of Qingdao City. It is 30 m wide and 60 m above the sea surface, and the road surface consists of asphalt, which has markedly different spectral characteristics than the sea surface [24].

## 3. Data and Methods

GOCI monitors the sea surface in eight spectral bands (Table 1, [25]). Rrs (a Lever 2P product) is a basic variable observed in all eight bands, from which other products such as chl-a and TSS (Level 2A products) are derived. GOCI provides multiple algorithms to retrieve such products in the GOCI Data Processing System (GDPS). In particular, the default algorithms to generate chl-a and TSS products in the JZB area from Rrs are called OC2 (Equation (1), [26]) and Yellow Sea Large Marine Ecosystem Ocean Colour Group algorithm (YOC, Equation (2), [27]), respectively. All of these products were downloaded from the Korea Ocean Satellite Centre website, then processed by GDPS.

$$chl_a = e_0 + 10^{e_1 + e_2 R + e_3 R^2 + e_4 R^3},$$
$$R = log_{10}\left(\frac{Rrs(490)}{Rrs(555)}\right), \tag{1}$$

$$TSS = 10^{\left(c_0 + c_1(R_{rs}(555) + R_{rs}(670)) - c_2\left(\frac{R_{rs}(490)}{R_{rs}(555)}\right)\right)}, \tag{2}$$

where $e_0$, $e_1$, $e_2$, $e_3$, $e_4$, $c_0$, $c_1$ and $c_2$ are given constants. According to these two algorithms, Rrs (here measured at wavelengths 490, 555 and 670 nm) is the only variable that determines chl-a and TSS, and was thus used together with these products to determine the potential impact of the bridge.

**Table 1.** GOCI spectral bands [25].

| Band | Centre (nm) | Band-Width (nm) | Main Purpose |
|:---:|:---:|:---:|:---|
| 1 | 412 | 20 | Yellow substance and turbidity extraction |
| 2 | 443 | 20 | Chlorophyll absorption maximum |
| 3 | 490 | 20 | Chlorophyll and other pigments |
| 4 | 555 | 20 | Turbidity, suspended sediment |
| 5 | 660 | 20 | Baseline of fluorescence signal, chlorophyll, suspended sediment |
| 6 | 680 | 10 | Atmospheric correction, fluorescence signal |
| 7 | 745 | 20 | Atmospheric correction, baseline of fluorescence signal |
| 8 | 865 | 40 | Aerosol optical thickness, vegetation, water vapour reference over the ocean |

The normalised difference water index (NDWI), used here to determine the potential impact of bridges, serves to distinguish open water from land in remote-sensing images [28], and is given by:

$$NDWI = \frac{Green - NIR}{Green + NIR}, \tag{3}$$

where *Green* and *NIR* are the radiation in the green and near-infrared bands, respectively. NDWI is calculated in this study from Rrs band 4 (green) and band 8 (NIR) (Table 1).

## 4. Results and Discussion

The spatial distributions of chl-a and TSS for each GOCI pixel are shown in Figure 2 for three different cloud-free days (13 September and 4 November 2017, and 9 March 2018) in order to gain a first insight into possible trends. The chl-a values at the JZB Bridge appeared significantly higher than those in the adjacent water area (Figure 2a–c). TSS values at the JZB Bridge were also slightly higher on 9 March 2018 (Figure 2f), but these differences were less evident on 4 November and 13 September 2017 (Figure 2d,e). This first look at the GOCI products suggests that the JZB Bridge may affect GOCI chl-a and TSS values.

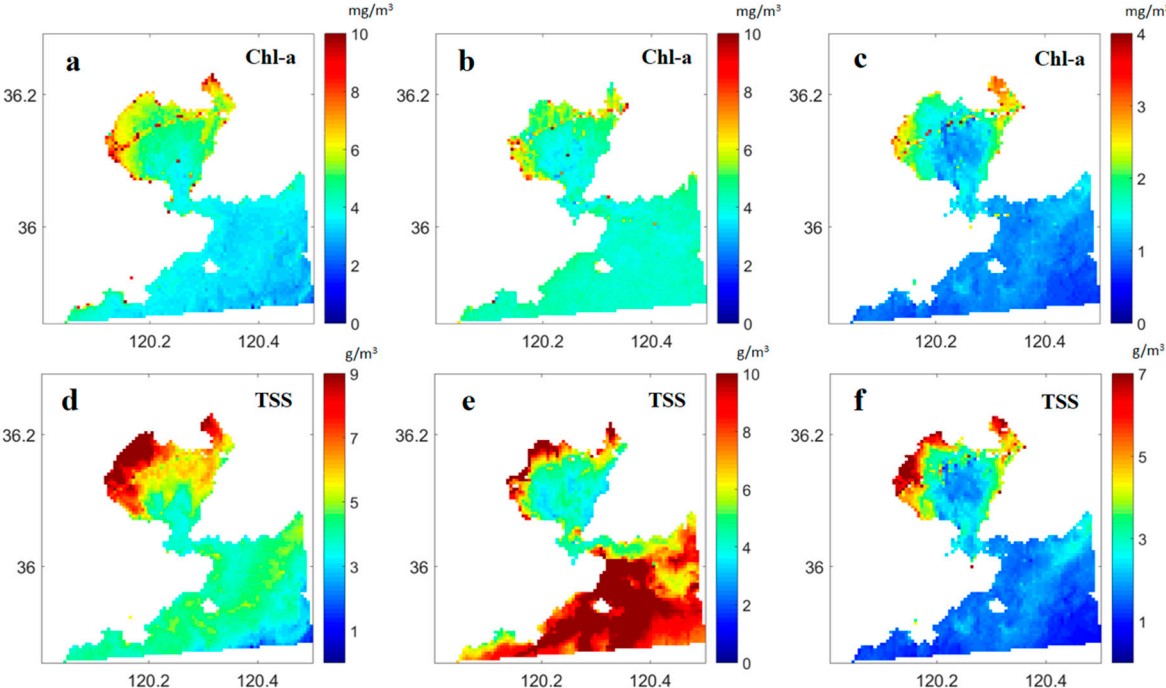

**Figure 2.** Distributions in JZB of concentrations of chlorophyll-a (**a**–**c**) and total suspended sediment (**d**–**f**) for each GOCI pixel on 13 September, 2017 (**a**,**d**), 4 November, 2017 (**b**,**e**) and 9 March, 2018 (**c**,**f**).

In order to determine whether these parameters were actually higher around the bridge area or if there was contamination of pixels by the bridge, we calculated the NDWI in the JZB area (Equation (3)) to identify land and water from the GOCI data (Figure 1c). The JZB Bridge is classified as land according to the NDWI results, which demonstrates that the reflection from the bridge surface affected the reflectance data for the pixels containing the bridge. The distributions of Rrs in the eight GOCI bands also illustrate this phenomenon. Figure 3 shows the spatial distributions of Rrs for each GOCI pixel and each band on 13 September 2017, a cloud-free day. The bridge's presence decreased the Rrs values in all the bands compared with the adjacent water area, even though the area of the JZB Bridge is only about 6% of the area covered by a GOCI pixel (500 m × 500 m). Contamination of all the Rrs bands implies a contamination of all the derived products. It is noteworthy that GOCI did not provide any warning about the possible contamination. These GOCI products are, however, regularly updated in a regional ocean database (JZB database, [29]), where contaminated data in the bridge area are flagged as land after a sea/land check during the data quality control, in order to warn users about abnormal data.

However, the bridge contamination was not evident in the lower-resolution (1000 m) Moderate Resolution Imaging Spectroradiometer (MODIS) data for chl-a in JZB on the same days—13 September 2017 and 9 March 2018 (data for 4 November 2017 were unavailable, figures not shown), presumably because the area of the bridge is too small (3%) relative to the area covered by a pixel. More comparative data are required to confirm this.

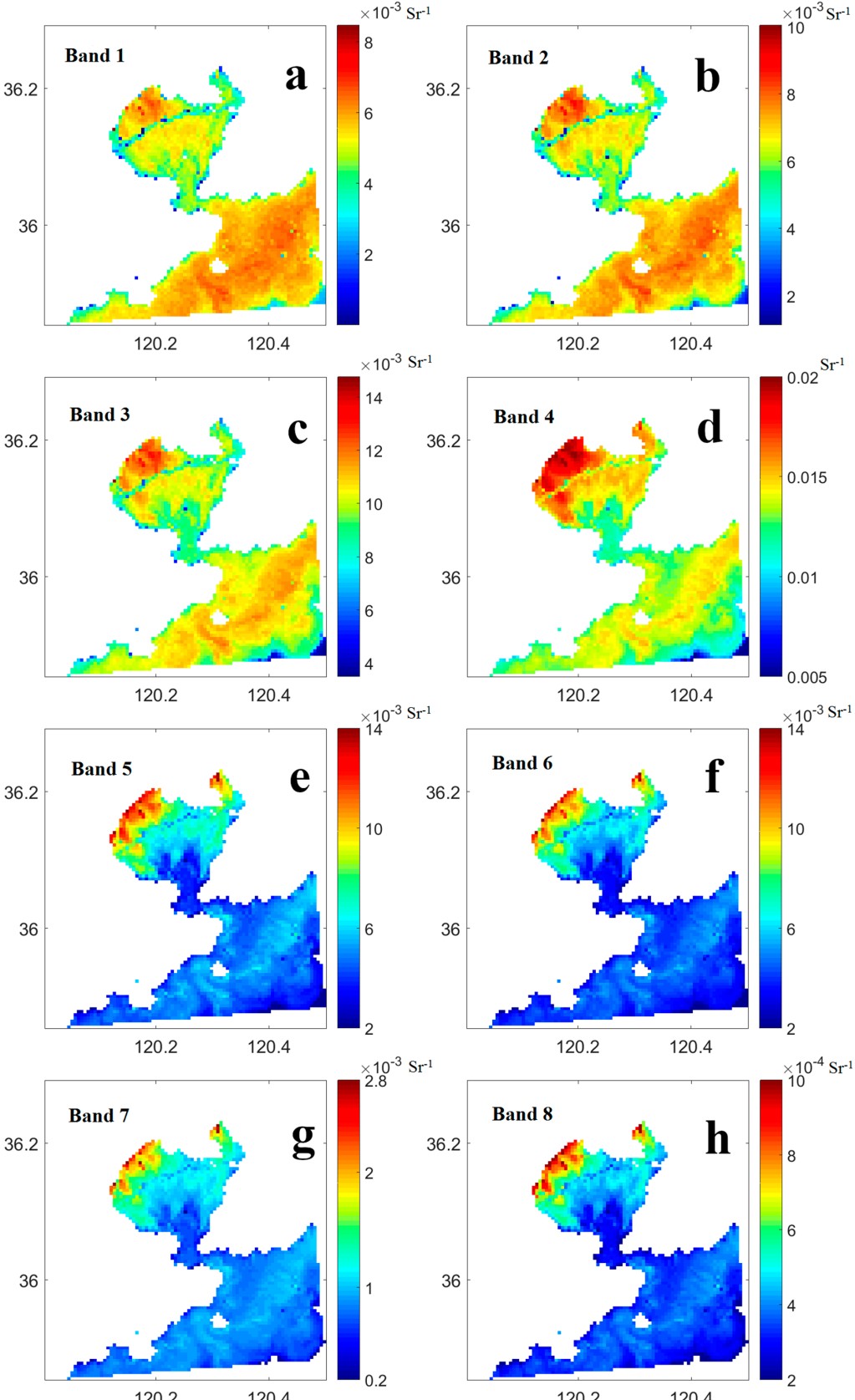

**Figure 3.** Distributions of Rrs for each pixel in the eight GOCI bands (**a**–**h**) following the order in Table 1) on 13 Sep 2017 in JZB.

The GOCI data displayed above were all selected based on their observed time (12:16 pm in local time) when the Sun nearly reached its highest point in the day, in order to minimize the potential impact of the bridge shadow. However, to check the potential impact of shadows on the pixel contamination, we verified the sun's position regarding the zenith angles of GOCI ($Z_G$) and the sun ($Z_S$), both provided by GOCI for each pixel. Since JZB is located north of the Tropic of Cancer, the nadirs of the sun and GOCI are both in the south side of the JZB Bridge. There are two possible scenarios for the bridge shadow being observed by GOCI (Figure 4): (1) when $Z_G$ is larger than $Z_S$, the bridge shadow cannot be observed (Figure 4a); (2) otherwise, the bridge shadow can be partly viewed (Figure 4b). The data observed at 12:16 p.m. on 13 September 2017 and 9 March 2018 correspond to the first case (shadow was not observed and had no effects) whereas the data observed at 12:16 p.m. on 4 November 2017 correspond to the second case (shadow was partially observed and may have influence). Whether or not the shadow was visible, the pixel contamination existed consistently (Figures 3 and 4). Consequently, we can conclude that the bridge shadow was not the main source of contamination.

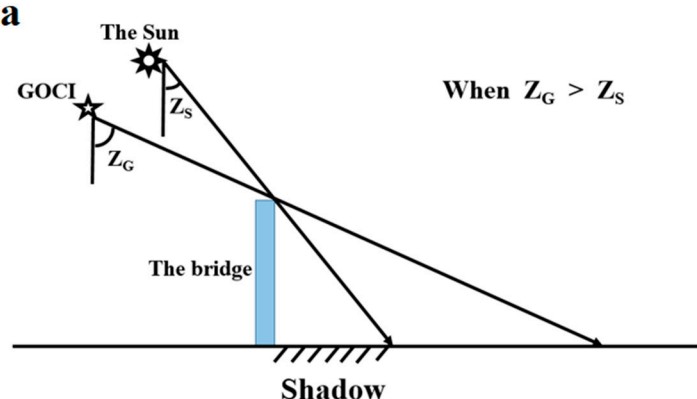

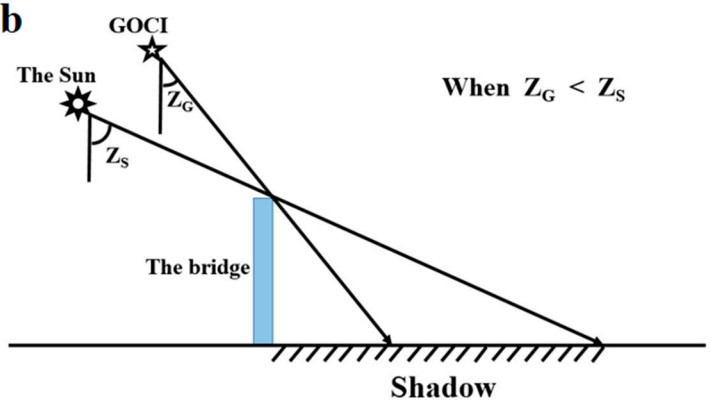

**Figure 4.** The bridge shadow from GOCI viewed when the zenith angle of GOCI ($Z_G$) is: (**a**) larger than the zenith angle of the sun ($Z_S$); and (**b**) smaller than $Z_S$.

To see if bridge contamination occurs more generally, the spatial distributions of chl-a and Rrs (band 4) at 10:16 a.m. (local time) on 27 July 2017 were plotted for Hangzhou Bay (HZB, Eastern China, Figure 5). HZB also has a cross-sea bridge of 35.7 km in length (Figure 5a), and is therefore another good example to test the impact of coastal infrastructure on ocean colour remote sensing. Both chl-a and Rrs values near the bridge were significantly different from those in the adjacent water, as for JZB. This suggests that the pixel contamination of remote-sensing products is common in coastal areas with such infrastructure, and should be considered in data post-processing.

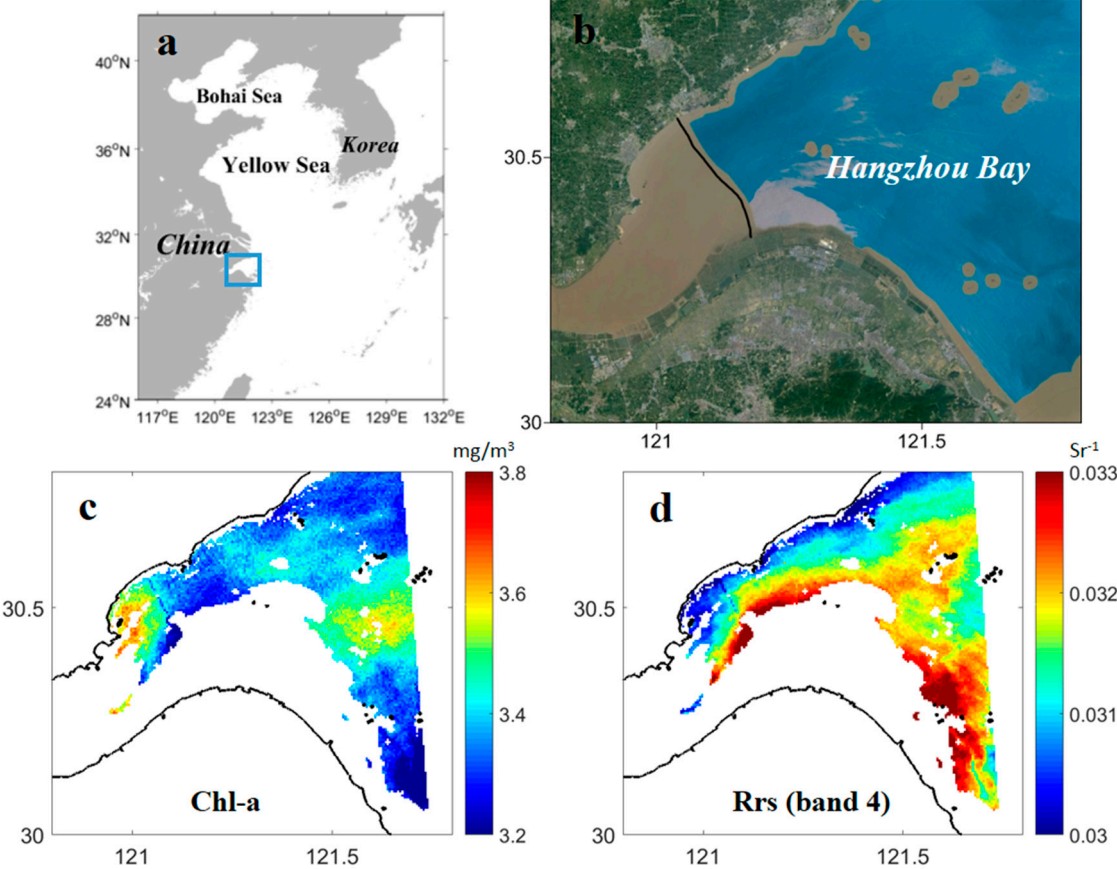

**Figure 5.** (**a**) Location of HZB on the China Sea coast. (**b**) Satellite image of HZB from Google Earth (provided by DigitalGlobal); the black line is the HZB Bridge. Distributions of (**c**) chl-a and (**d**) Rrs in band 4 for each GOCI pixel on 27 July 2017 in HZB, with the shoreline in black.

In conclusion, coastal infrastructure, such as cross-sea bridges, can contaminate ocean colour data from remote sensing due to the different spectral characteristics of the bridge and the seawater surface. We have demonstrated that the differences in chl-a and TSS values between the JZB Bridge and the adjacent water are caused by the bridge contamination of nearby pixels. Chl-a values were also contaminated by the bridge in HZB. Therefore, we conclude that coastal infrastructure has an impact on ocean colour remote sensing, but the impact is sensitive to the pixel resolution supported by various datasets; it was evident in the high-resolution GOCI data but not in the lower-resolution MODIS data. It is recommended that the pixels containing infrastructure in studies using higher-resolution satellite products should be discarded. This short communication comprises the first evaluation of the impacts of coastal infrastructure on ocean colour remote sensing, and will help to improve the quality of remote-sensing products and therefore the accuracy of product applications, such as water quality evaluations or model validations.

**Author Contributions:** Conceptualization, X.H.W.; methodology, X.H.W., I.J.-R. and Y.Y.; validation, Y.Y.; formal analysis, X.H.W., I.J.-R. and Y.Y.; data curation, Y.Y. Writing—original draft preparation, Y.Y.; writing—review and editing, I.J.-R., X.H.W. and Y.Y.; visualization, Y.Y.; supervision, X.H.W. and I.J.-R.

**Funding:** This research received no external funding.

**Acknowledgments:** This short communication benefited from editorial review by Peter McIntyre from UNSW Canberra. Yuan Yuan is supported by the China Scholarship Council and a UNSW Canberra Top-up Scholarship. This is publication No. 68 of the Sino-Australian Research Centre for Coastal Management.

**Conflicts of Interest:** The authors declare no conflict of interest.

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
