# Peer review of "Impact of Coastal Infrastructure on Ocean Colour Remote Sensing: A Case Study in Jiaozhou Bay, China"

_remotesensing, doi:10.3390/rs11080946_

Round 1

Reviewer 1 Report

The authors present a generally well written manuscript on how infrastructure can alter remotely-sensed ocean color data and should be factored into post-processing analyses. With coastal development resulting in heavy modifications to our estuaries it is certainly valid to ensure studies are factoring in potential contamination from infrastructure on remotely-sensed products.

General comments:

The introduction does not mention any other studies that have evidenced color contamination from infrastructure. Is this the first? Or is this the first with ocean color? Could there be some terrestrial parallels that could be referenced?

I would like to see more information about the different aspects of the bridges, particularly more detailed dimensions. The authors mention the length of the JZB bridge in the study area and then the width later in the results, but how tall is the bridge? Along this line of thought, it might be helpful to note the time of day that the GOCI data are obtained. Depending on the height of the bridge and time of day, could shadow also be making the bridge (which is small compared to the GOCI pixel) more pronounced in the ocean color data?

Is there any indication at what scale this contamination could cause significant impact to various datasets?  Seems like at a coarse scale it may be lost in overall noise, but there are certainly instances where it would have a significant detrimental impact.

Figures: I would find it helpful to have more detailed axes labels, particularly with specific variables and units so as not to have to depend on the caption for some of that information.

Author Response

Point 1: The introduction does not mention any other studies that have evidenced color contamination from infrastructure. Is this the first? Or is this the first with ocean color? Could there be some terrestrial parallels that could be referenced?

Response 1: Yes, it is an inspired comment.

It is the first study on the ocean colour. We have mentioned it at Line 63, in the paragraph that describes the aim of this work. And we also emphasized it in conclusion at Line 184.

We have added several sentences between Line 52 and 56 to compare with terrestrial remote sensing and included terrestrial parallels.

Point 2: I would like to see more information about the different aspects of the bridges, particularly more detailed dimensions. The authors mention the length of the JZB bridge in the study area and then the width later in the results, but how tall is the bridge? Along this line of thought, it might be helpful to note the time of day that the GOCI data are obtained. Depending on the height of the bridge and time of day, could shadow also be making the bridge (which is small compared to the GOCI pixel) more pronounced in the ocean color data?

Response 2: This is an interesting point. We have added the width (30 m) and height (60m) of the bridge in the Introduction section (Lines 93 and 94), and mentioned the exact time of data observation: at 12: 16 pm in local time, when the Sun nearly reached its highest point in the day, in order to minimize the potential impact of the bridge shadow (Lines 148 and 165). We have also added a paragraph and a figure to discuss about the potential impact of the bridge shadow between Line 148 and 163, and concluded that the bridge shadow is not the main source of contamination.

Point 3: Is there any indication at what scale this contamination could cause significant impact to various datasets?  Seems like at a coarse scale it may be lost in overall noise, but there are certainly instances where it would have a significant detrimental impact.

Response 3: We checked MODIS chl-a data (1000 m × 1000 m in resolution), and found no trend of bridge contamination. We have added such discussion from Line 135 to Line 140. GOCI has the finest resolution for ocean colour remote sensing in JZB, so we cannot find another instance about the infrastructure impact.

Point 4: Figures: I would find it helpful to have more detailed axes labels, particularly with specific variables and units so as not to have to depend on the caption for some of that information.

Response 4: We have improved the Figs. 2, 3, 5, by adding the specific variables and units in every sub-figures.

Reviewer 2 Report

I would expect a bit more discussion on water current regimes in the Bay. Such works have been published. This may be important regarding the distribution of chl-a and TSS in the Bay. Also, slightly more extended conclusions, with a short discussion on the importance of this research and perhaps some recommendations for future activities regarding such regions.

Author Response

Point 1: I would expect a bit more discussion on water current regimes in the Bay. Such works have been published. This may be important regarding the distribution of chl-a and TSS in the Bay. 

Response 1: We have added related sentences between Line 80 and 86, including currents and its magnitude in JZB, also with residual currents characters, and the wind effects.

Point 2: Also, slightly more extended conclusions, with a short discussion on the importance of this research and perhaps some recommendations for future activities regarding such regions.

Response 2: We have improved the conclusion following this suggestion, by adding significance and recommendations. 

Reviewer 3 Report

Review of article entitled “Impact of coastal infrastructure on ocean colour remote sensing: a case study in Jiaozhou Bay, China” 

The article deals with the effects that a large infrastructure might produce on ocean color products in the middle of a regional sea, using as an example the Jiaozhou Bay Bridge, located in the Jiaozhou Bay and also reinforce the hypothesis with another case of study, the Hangzhou Bay. To achieve the objective, the authors performed a visual analysis of standard chl-a, TSS and Reflectance products derived from GOCI. From the Rrs and Chl-a prodcuts pixel contamination can be easily observed.  

The article is well written, and easy to follow and understand. The figures are very clear and they are in agreement with the text. 

Even the results are important for the future remote sensing studies conducted in the study area, the methodology, the results and discussion are too simple and shallow and might not be appropriate to be published as a technical note.  

Specific Comments: 

- Line 27-34: please add some referencesAlso note that other products such as CDOM and TSS between others, are not new. I believe this paragraph should be re-writtenfocusing on which parameters could be derived from remote sensing which could be used as a proxy of biogeochemical parameters and as a result to monitor water quality and processes. 

- Line 38 and 39. You mentioned that parameters along Korean coast were validated, but you do not mention which algorithms and the retrieval they had. Please add that information.  Notice that OC2 presented no good results in Korean coastal waters (Moon et al., 2012). So, it was validated but the results indicated that that algorithm is not good for Korean coastal waters, and probably also for the Jiaozhou bay. Please, be clear with those aspects. 

-In the method section: I don’t see the point of adding the formulae for chl-a and tss algorithms, I believe with mentioning the name and the reference of the algorithm is OK. 

-In results and discussion section: I would suggest presenting  the Figure of Rrs first, and then the one of chl-a and tss, since as the Rrs is affected by the bridge, then is supposed that the chl-a and tss products will be affected too, since they are based on the Rrs product. 

Author Response

Point 1: Even the results are important for the future remote sensing studies conducted in the study area, the methodology, the results and discussion are too simple and shallow and might not be appropriate to be published as a technical note.  

Response 1: We have changed the manuscript type from Technical Note to Short Communication, following the suggestion by the editor. We have demonstrated that the methodology, the results and discussion are robust and appropriate to reach the objective. The motivation to write such a communication, is that we have encountered a practical problem in using the GOCI product as non-remote sensing specialists. When we plotted the chl-a distributions in JZB, at first the abnormal distribution at the bridge gave us a wrong signal that the bridge may have hydrodynamical or biological impacts on chl-a. Then we checked the Rrs and NDWI, and found that it is pixel contamination. We hope other non-specialist users can avoid similar problems. That is our motivation, and we believe this short communication has its significance.

Point 2: Line 27-34: please add some references. Also note that other products such as CDOM and TSS between others, are not new. I believe this paragraph should be re-written, focusing on which parameters could be derived from remote sensing which could be used as a proxy of biogeochemical parameters and as a result to monitor water quality and processes.

Response 2: We have re-written this paragraph following your suggestion (Lines 33 to 40).

Point 3: You mentioned that parameters along Korean coast were validated, but you do not mention which algorithms and the retrieval they had. Please add that information.  Notice that OC2 presented no good results in Korean coastal waters (Moon et al., 2012). So, it was validated but the results indicated that that algorithm is not good for Korean coastal waters, and probably also for the Jiaozhou bay. Please, be clear with those aspects. 

Response 3: We have improved this part following your suggestion (Lines 45 and 46). Although the OC2 is not good, the Rrs is reasonable. So the following discussion about bridge impacts is still appropriate.

Point 4: In the method section: I don’t see the point of adding the formulae for chl-a and tss algorithms, I believe with mentioning the name and the reference of the algorithm is OK. 

Response 4: We prefer to keep the formula in order to show the contribution of Rrs in each procduct. This can help non-specialists in remote sensing to understand why we discuss the contamination of Rrs.

Point 5: In results and discussion section: I would suggest presenting  the Figure of Rrs first, and then the one of chl-a and tss, since as the Rrs is affected by the bridge, then is supposed that the chl-a and tss products will be affected too, since they are based on the Rrs product.  

Response 5: Yes, this order is another possibility. However, we prefer to show first the problem and then the diagnostic. We think that this order can help users who have the same problem, a strange trend in ocean colour products around an infrastructure, to understand the motivation of our analysis. We prefer to keep the original order.

Round 2

Reviewer 3 Report

After the revision the article has improved. I believe it is certainly more appropriate to present it as a “short communication” instead as a “technical note”.

Minor comments:

Line 31-36: please add some references.